# The Effectiveness of Massage Therapy for Improving Sequelae in Post-Stroke Survivors. A Systematic Review and Meta-Analysis

**DOI:** 10.3390/ijerph18094424

**Published:** 2021-04-21

**Authors:** Rosa Cabanas-Valdés, Jordi Calvo-Sanz, Pol Serra-Llobet, Joana Alcoba-Kait, Vanessa González-Rueda, Pere Ramón Rodríguez-Rubio

**Affiliations:** 1Physiotherapy Department, Faculty of Medicine and Health Sciences, Universitat Internacional de Catalunya, 08195 Barcelona, Spain; polserra@uic.es (P.S.-L.); jowyalka@uic.es (J.A.-K.); vgonzalez@uic.es (V.G.-R.); prodriguez@uic.es (P.R.R.-R.); 2Physiotherapy Department, School of Health Sciences, Tecno Campus, Mataró-Pompeu Fabra University (TCM-UPF), 08302 Barcelona, Spain; jcalvo@tecnocampus.cat; 3Hospital Asepeyo Sant Cugat del Vallès, 08174 Barcelona, Spain; 4CENAC, 08028 Barcelona, Spain; 5Fundació Institut Universitari per a la Recerca a l’Atenció Primaria de Salut Jordi Gol i Gurina, 08007 Barcelona, Spain

**Keywords:** massage therapy, stroke, motor function, spasticity, Tuina, soft manipulation

## Abstract

Objective: To assess the effect of therapeutic massage for improving sequelae in stroke survivors. Methods: A systematic review of the nine medical databases from January 1961 to December 2020 was carried out. The bibliography was screened to identify randomized controlled clinical trials (RCTs). Two reviewers independently screened references, selected relevant studies, extracted data and assessed the risk of bias using the PEDro scale. The primary outcome was upper and lower limb motor function and spasticity. Results: A total of 3196 studies were identified and 18 RCT were finally included (1989 individuals). A meta-analysis of RCTs in the comparison of Chinese massage (Tuina) plus conventional physiotherapy versus conventional physiotherapy was performed. The mean difference (MD) in the subacute stage on upper limb motor-function using the Fugl Meyer Assessment was 2.75; (95% confidence interval (CI) from 0.97 to 4.53, *p* = 0.002, I^2^ = 36%). The MD on upper limb spasticity using modified Ashworth scale was −0.15; (95% CI from −0.24 to −0.06, *p <* 0.02, I^2^ = 0%).The MD on lower limb spasticity was −0.59; (95% CI from −0.78 to −0.40, *p <* 0.001, I^2^ = 0%) in the endpoint. Conclusions: Therapeutic massage, especially Tuina, in addition to conventional therapy is effective for improving motor function and for reducing spasticity in stroke survivors.

## 1. Introduction

Stroke is a leading cause of long-term adult disability. The individuals that have suffered from a stroke present various upper motor neuron syndrome, signs and symptoms, including weakness, spasticity, lack of coordination and agonist antagonist co-contraction, with up to 50% of survivors being chronically disabled [1]. They involve together in impairments and functional problems that can lead to costly complications. [2]. Physical therapy may contribute to the improvement of disabilities and quality of life in these individuals [3].

Manual therapeutic massage is the most applied type of passive physical therapy and it is one of the oldest forms of medicine known to humanity, having been practiced worldwide since ancient times [4]. All massage manipulations introduce mechanical forces into the soft tissues by means of “mechanotransduction” [5]. Massage can increase muscle mass temperature and blood flow, and this might help to increase muscle compliance and minimize muscle stiffness [6]. There are several kinds of therapeutic massage. The most common type of massage in the Western world is Swedish massage. This is one of the common treatments for provide optimal performance among athletes, and is based on the Western concepts of anatomy and physiology [7]. It involves the systematic application of manual pressure and the movement of soft tissue, with rhythmical pressure and stroking to obtain or maintain health [8]. Another type is Chinese massage (Tuina). This involves various strokes, shaking stretching and joint movement along energy channels to balance the body’s energy, as well as physical and emotional system [9]. Indian massage (Dalk) includes the manipulation of body tissues with the hands. In Unani medicine, Dalk is based on the principle of *tanqiyah* (expulsion) and *imāla* (diversion) [10]. Finally, Thai massage is a form of deep massage involving brief sustained pressure on the muscles. Pressure point massage along the body’s hypothesized 10 major energy channels or *Sen Sib* is believed to release blocked energy and to increase awareness and vitality [11].

According to research, therapeutic massage can effectively improve mood, create a feeling of pleasure, and reduce the occurrence of major adverse events and injuries [12] in cancer patients [13], in para-athletes [14], in some neurological conditions such as Parkinson’s disease [15], in dementia sufferers [16] and in post-stroke constipation [17]. However, the scientific evidence that supports the effectiveness of therapeutic massage in stroke survivors is limited. Because of the trends of previous studies on therapeutic massage were reported heterogeneously, we performed a meta-analysis and systematic review for evidence-based treatment. The purpose of this systematic review is therefore to analyze the evidence for therapeutic massage for improving motor function, spasticity, activities of daily living, anxiety, pain, balance, gait, stroke disability and quality of life in adult stroke survivors.

## 2. Materials and Methods

### 2.1. Protocol and Registration

We conducted a systematic review of the scientific literature to assess the effect of therapeutic massage interventions in stroke survivors. The PROSPERO (register of systematic reviews) number was CRD42020178942. The guidelines in the Cochrane Handbook for Systematic Reviews of Interventions Version 6 [18] and PRISMA [19] statement were followed.

### 2.2. Eligibility Criteria

Studies in Chinese, Spanish, French, Italian, Portuguese and English published between January 1961 and December 2020. Since only RCTs were to be analysed all other forms of literature were excluded as comments, reviews, observational studies, books, poster/oral abstract communications, case reports, non-randomized studies, protocols, systematic reviews and practice guidelines.

Therapeutic massage was compared to no treatment, sham treatment or active treatment. We defined therapeutic massage in this revision as “a patterned and purposeful soft-tissue manipulation accomplished by the use of digits, hands, forearms, elbows, knees and/or feet, with or without the use of emollients, liniments, heat and cold with the objective of therapeutic change” [20].

### 2.3. Data Items

The search strategy was designed using the PICO framework. (P) Adult post-stroke survivors, (I) receiving therapeutic massage alone, or combined with another rehabilitation approach or conventional physiotherapy, (C) compared to patients receiving another rehabilitation approach or conventional physiotherapy and (O) changes in the analysis of upper/lower limbs motor function, spasticity, activities of daily living, anxiety, pain, balance, gait, stroke disability and quality of life with or without a follow-up assessment.

The primary outcomes were upper/lower motor function and spasticity. The secondary outcomes were activities of daily living, pain, anxiety, balance, gait, stroke disability and quality of life.

### 2.4. Search

A computerized search strategy of the following databases was performed: Medline/PubMed, Cochrane Central Register of Controlled Trials (CENTRAL), Physiotherapy Evidence Database (PEDro), Scielo, Tripdatabase, Web of Science, Scopus, CINHAL and Epistemonikos. A manual search was also performed. We used the Medline search strategy, and adapted it to other databases (Appendix B).

Searches for eligible articles and data extraction were conducted independently by three authors (VG-R/JC-S/RC-V). Proquest Refworks discarded duplicate articles, and the remaining studies were analyzed for their appropriateness. Selection was initially based on the title or abstract, and subsequently on the full text of the articles. They were thoroughly checked to confirm the selection criteria. We analyzed whether the studies included followed the template for the intervention’s description and the replication checklist (TIDieR) [21]. The following data were extracted: patient stroke characteristics, number of participants in each group, type of massage, co-interventions, zone of massage application, number of sessions, comparisons, outcome measures and tool used, follow up and main results. Conventional physiotherapy interventions were described.

### 2.5. Risk of Bias in Individual Studies

The risk of bias assessment was assessed by two authors (J.A.-K. and P.S.-L.) using the PEDro scale [22]. In cases of doubt or disagreement, a discussion took place with a third reviewer (R.C.-V.) until a consensus was reached. The PEDro is an 11-item scale, in which the first item relates to external validity (not used to calculate the total score) and the other 10 items assess the internal validity of a study. The higher the score, the greater the study’s risk of bias as assessed by the following cut-points: 9–10: excellent; 6–8: good; 4–5: fair; <4: poor [23].

### 2.6. Synthesis of Results

Data from the studies were summarized narratively using text and tables. Studies were grouped by comparator. A meta-analysis was performed whenever possible. The treatment effect sizes were calculated using the Revman 5.3 software package [24], based on the mean scores and standard deviations of the studies. Post-intervention effects were analyzed by calculating the change between the baseline and the immediate post-intervention assessment and persisting effects by computing the change between the baseline and the final follow-up assessment. These changes were compared between groups. When the outcomes were continuous and measured in the same unit, a mean difference was used; otherwise, a standardized mean difference was used. The effect size was categorized as 0.2, 0.5, 0.8 and 1.3, which were considered small, medium, large and very large, respectively. Funnel plots were used to illustrate the risk of publication bias [25].

The heterogeneity was assessed visually by means of forest plots and by reporting the I^2^ statistic. The I^2^ statistic describes the percentage of total variation across studies that are attributable to heterogeneity rather than chance. A value greater than 25% is considered to reflect low heterogeneity, 50% moderate, and 75% high heterogeneity [26]. The fixed effect model was applied by default, and the random-effect model was used in cases of substantial heterogeneity [27]. When there was insufficient data for quantitative analysis, the review only represents and summarizes the evidence. Missing data was requested by contacting the corresponding author.

## 3. Results

### 3.1. Study Selection

The PRISMA diagram (Figure 1) summarizes the results of the scientific literature search and 18 randomized controlled trials were included [28,29,30,31,32,33,34,35,36,37,38,39,40,41,42,43,44,45]. No author responded when contacted for additional information.

### 3.2. Risk of Bias within Studies

The mean PEDro score assessing the risk of bias was 6.3 points (range 4–9) from 10 criteria (Appendix A), indicating a good score. Only one study [35] with 9 points was found, because is difficult to blind the therapist and patients in a physical therapy intervention.

### 3.3. Study Characteristics

An overview of the studies included and the patients’ characteristics was provided (Table 1). The total population analyzed included 1989 individuals, of whom 1273 were male and 683 were female. Ahmed (2015) did not specify the gender distribution, 1057 had suffered from an ischemic stroke and 511 were hemorrhagic, and five studies did not report this information. There were 439 patients with left side hemiparesis, 420 had right side hemiparesis and it was bilateral in 63 individuals and not specified in eleven studies. The age of the participants ranged from 32 to 86 years old and the average of most studies was 60 years. Four studies were multicenter. Most of the studies performed their intervention in the subacute stroke phase (≤3 months), while only three studies recruited a mixture of chronic and subacute phase. Three authors did not specify which phase.

Regarding outcomes (Table 2) eleven studies assessed motor function and ten of them used the Fugl Meyer Assessment [46]. Six authors evaluated spasticity, and all authors used the modified Ashworth Scale [47]. Most of the studies performed their intervention in the subacute stroke phase (≤3 months), while only four studies [30,32,36,38] recruited a mixture of chronic and subacute phase. Three authors did not specify which phase. Only four studies [35,36,42,45] performed a follow-up of 3 months, one had a follow-up of 6 months, and one had a follow-up of 3 days. Wang (2019) was the study that recruited the most patients, with 397.

As regards as therapeutic massage only four studies [28,30,33,41] performed massage alone in the experimental group. Twelve studies performed Tuina massage, six of them combined it with conventional physiotherapy, three performed Tuina with acupuncture, two performed Tuina with acupuncture plus conventional physiotherapy, one performed Tuina in combination with medicinal herbs. Three studies performed Dalk massage, one performed slow-stroke back massage; another performed Thai massage and one performed Swedish massage in combination with footbath and conventional physiotherapy. The number of massage sessions ranged from 7 to 40, the frequency was usually once a day and the duration of treatment ranged from 1 to 8 weeks, with a massage time ranging from 10 to 60 min. Most of the studies involved daily sessions except for Pan (2011) and Han (2015), which involved 2 sessions per day lasting 40–50 min per massage session (Appendix A).

### 3.4. Efficacy of Therapeutic Massage

Two comparisons were performed: (1) Tuina massage plus conventional physiotherapy versus conventional physiotherapy; (2) Tuina massage plus acupuncture versus conventional physiotherapy. A meta-analysis was performed for motor function, spasticity, and activities of daily living.

### 3.5. Upper/Lower Limbs Motor Function

For the first comparison, the mean difference (MD) was performed using Meyer Assessment. It was 1.74; (95% confidence interval (CI) from −0.30 to 3.77, *p* = 0.09, I^2^ = 66%) in the endpoint and 2.90; (95% CI from −048 to 6.28, *p* = 0.09, I^2^ = 60%) in the follow up (Figure 2). A sub-analysis for upper limb motor function was performed in subacute stage (≤3 months) the MD was 2.75;(95% CI from 0.97 to 4.53, *p* < 0.02, I^2^ = 36%) in the endpoint (Figure 3).

For the second comparison a meta-analysis was performed. The MD was 1.03; (95% CI from −0.03 to 2.03, *p* = 0.04, I^2^ = 0%) in the follow up (Figure 4).

### 3.6. Spasticity

The first comparison was performed. The MD for upper limb spasticity assessed by Modified Ashworth Scale was −0.14; (95% CI from −0.21 to −0.07, *p <* 0.02) in the endpoint and the MD was −0.32; (95% CI from −0.41 to -0.23, *p <* 0.001) in the long-term and the heterogeneity was low at 0% in both case (Figure 5). A sub-analysis was performed in subacute stage for upper limb spasticity the MD was −0.15; (95% CI from −0.24 to −0.06, *p <* 0.02, I^2^ = 0%) in the end point and −0.36; (95% CI from −0.47 to −0.25, *p* < 0.001, I^2^ = 0%) in the 3-month follow-up (Figure 6).

### 3.7. Activities of Daily Living

The first comparison was performed. The MD for activities of daily living at the endpoint evaluated by the Barthel index was 1.91; (95% CI from −0.98 to 4.80, *p* = 0.20) and in the long-term it was 0.26 (95% CI from −3.19 to 3.71, *p* = 0.88) (Figure 7).

### 3.8. Anxiety and Stress

Anxiety and stress were evaluated only in three studies with different interventions, involving Swedish, Thai and slow-stroke back massage, and all had positive results.

### 3.9. Pain

Tuina massage was effective at reducing shoulder-hand pain. One study Yang (2011) also performed it in combination with conventional physiotherapy, while another Li (2012) performed it in combination with electrical acupuncture.

### 3.10. Gait and Balance

Regarding gait, Pan HP (2011) found that Tuina massage improves it and Zarnigar (2012) improves it with Unani exercises. Wu (2013) observed that balance with Tuina massage plus balance training was statistical significant better than in the control group.

### 3.11. Adverse Events

No adverse events were reported during therapeutic massage treatment. All adverse events were considered unrelated to the rehabilitation techniques. The drop-out and loss of patients during the follow-up were unrelated to the study intervention.

### 3.12. Risk of Bias Publication

All the funnel plots were symmetrical, and as such, publication bias is low, the following are available online (Appendix A).

## 4. Discussion

The main findings of this systematic review and meta-analysis are that therapeutic Chinese massage (Tuina) combined with conventional physiotherapy is an effective method to improve motor function and to reduce spasticity in stroke survivors, especially in subacute stage. The combination of Tuina massage plus acupuncture also improves the symptoms. The results of this review are important, because the therapeutic massage intervention was mostly performed in the subacute stage of the stroke, with positive results in upper/lower limbs motor function. Recovery of upper limbs function remains a major scientific, clinical and patient priority [48].

Surprisingly this review only found one study that used the Swedish massage as an intervention. It was used to decrease anxiety. This was unexpected to us, because Swedish massage is today the most popular and best-known type of massage in the Western world [49]. We expected to find more studies as other authors had used Swedish massage to improve spasticity and motor function in multiple sclerosis [50,51] and cerebral palsy [52,53]. Scientific publications in Europe, America, Africa or Australia were not found, they were only found in Asian countries, and especially in China.

The upper and lower limbs motor function was the outcome most evaluated, followed by spasticity the two outcomes are linked [54]. Motor impairments in stroke survivors can be described by a cycle of overactivity-contracture-overactivity evolving in parallel with the continuum of paresis-disuse-paresis. Both cycles must be disrupted to optimized motor recovery and function [55]. In fact, a more complete restoration of motor function is achieved when spasticity is absent [56]. It is important to reduce spasticity before the patient performs the voluntary movement in order to obtain a movement with some quality since this will influence the neuroplasticity of the individuals and their recovery [57]. According to several authors [54,58,59] the sensory system has an important role in spasticity mitigation and is the most important predictor for severe spasticity.

The mechanism behind elastic modulus changes in spastic muscle in stroke survivors is still under discussion. One possible hypothesis might be related to structural alterations in the muscle after a stroke. Shortened muscle fascicle length in the upper limb [60] and lower limb [61] has been observed. These results suggest that altered muscle morphology of the paretic muscle may contribute to abnormal muscle elastic properties during passive stretching [62].

As a result of damages to the motor cortex and its descending pathways and the subsequent unmasking of inhibition, there is evidence of upregulation of reticulospinal tract projections excitability on the contralesional side in stroke survivors [63]. Reticular nuclei receive sensory input from the periphery and neck proprioceptors. In addition to sensorimotor integration, the reticular formation also seems to play a role in preparation for a voluntary movement [64]. Therapeutic massage increases blood flow and parasympathetic activity, releases relaxation and stress hormones, and inhibits muscle tension, and neuromuscular excitability [65]. It could reduce the hyperexcitability of the reticulospinal tracts. The various types of therapeutic massage modalities could be most useful for the therapist to reduce muscle overactivity to enable other therapeutic interventions.

The results obtained for daily living activities, gait, balance, quality of life and stroke severity were inconclusive. The trend is positive when Tuina is used in addition to the conventional therapy or acupuncture. Regarding pain, our results are consistent with the literature available [66]. There is growing evidence to support the concept of an interactive network between the cutaneous nerves, the neuroendocrine axis and the immune system [67]. Therapeutic massage is reported to have several beneficial effects, including activation of the relaxation and growth response has been suggested to be mediated by oxytocin [68]. Stroke patients suffer from anxiety and massage could be helpful for relaxation to ease the patients’ suffering [69]. The results of this review show positive effects, as it reduced anxiety in stroke survivors. Surprisingly, no study evaluated the range of motion as an important aspect to take into account when reducing spasticity [70].

Functional magnetic resonance imaging data have suggested that moderate pressure massage with movement is represented in several brain regions, including the amygdala, the hypothalamus and the anterior cingulate cortex, which are all areas involved in stress and emotion regulation [13,71]. Findings from the whole-brain meta-analysis of right-hand tactile stimulation highlight the importance of taking bilateral activation into consideration, particularly in the secondary somatosensory cortex [72].

Most of the articles in this systematic review used Tuina massage for improving outcomes. This is one of the four main branches of traditional Chinese medicine. However, although its roots in China are ancient, it is still relatively new in the West [9]. Tuina massage was originated from China over 5000 years ago and is commonly known today as “the grandfather of all therapeutic massage therapies”. It follows the meridian theory and works on the organs, energy channels in muscle groups and points on the body using the same principles as acupuncture, except hands and fingers are used instead of needles [73]. It is combined with anatomical and pathological diagnosis in order to achieve dredging meridian, removes pathogenic factors and has a curative effect of a harmonic balance of Yin-Yang [74]. Tuina can act on the subcutaneous muscular layer; enhance local blood circulation, lymph circulation, tissue metabolism of the skin, can regulate physiological and pathological states, unblock meridians, and harmonize Qi (total life energy). In Chinese medicine, Qi disorder and Yin-Yang imbalance account for balance disturbances following stroke. It corrects an imbalance in the yin and yang and qi (energy) which when translated into Western medical terminology, can also been understood as the “modulation of the imbalance between parasympathetic and sympathetic activity”. Yin-Yang imbalance also contributes to upper and lower limb spasticity following stroke, manifesting as ‘flaccidity of Yang and spasm of Yin’ [75].

Tuina massage encompasses techniques as grasping, pressing, rolling, round rubbing, holding-twisting, rub rolling, pushing, kneading, rotating, shaking, wiping, vibrating, digital striking, knocking, chapping, pressing, acupressure, myofascial release, reflexology, stretching techniques and joint mobilizations applied to specific body points [76]. Tuina is a functional massage and it can input substantial proprioceptive sensory impulses to the central nervous system through muscle, tendon and joint motion [77].

No study reports adverse events, but massage therapies are not totally devoid of risks. The incidence of adverse events is unknown, but is probably low [78]. The massage itself does not increase a person’s risk of stroke, but some precautions need to be taken with certain individuals. If the individual has blood clots, there is a small chance they could be dislodged by massage. Individuals on blood thinner medication bruise more easily, so deep tissue massage should be avoided. Care should be taken around the neck area in the region of the carotid artery, but this should not be an issue with an experienced massage therapist.

Further investigations are required at both the experimental and clinical levels to compare therapeutic Chinese massage (Tuina) versus Swedish massage in stroke survivors. Surprisingly, therapeutic massage is not on the list of recommendations in Western stroke management guides. Rehabilitation therapy based on integrated Chinese and Western medicine could be effective for stroke survivors [79]. Finally, touch a patient has a therapeutic value and it has many benefits [80].

The current review has several limitations. First, the majority of patients in this review are Asian people. Second, only one Swedish massage article was found for stroke survivors. We do not know if Tuina massage will be effective in Western population.

## 5. Conclusions

The results of this systematic review suggest that therapeutic Chinese massage (Tuina) in addition to conventional physiotherapy is an effective non-invasive treatment for improving upper/lower limbs motor function and for reducing spasticity especially in the subacute stage of stroke.

## Figures and Tables

**Figure 1 ijerph-18-04424-f001:**
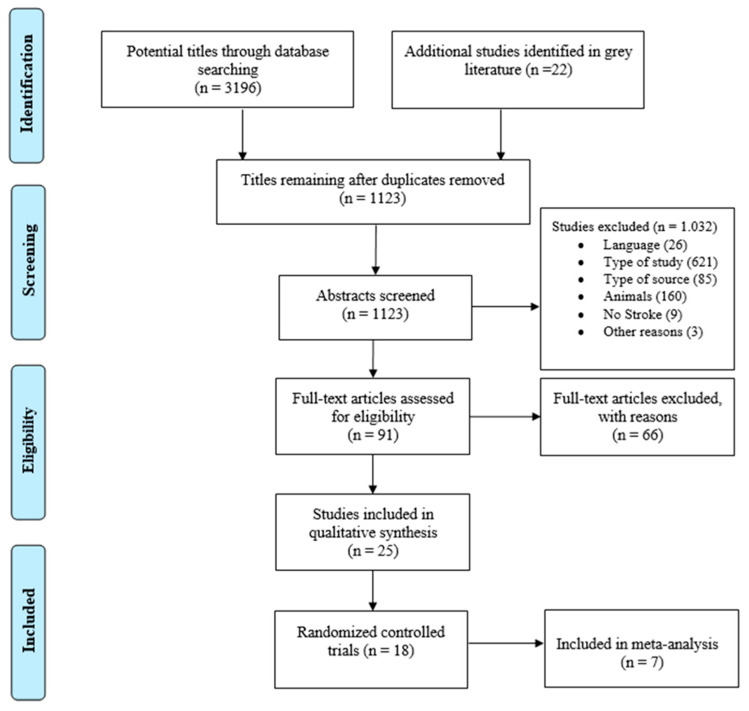
PRISMA diagram of the process used to identify studies.

**Figure 2 ijerph-18-04424-f002:**
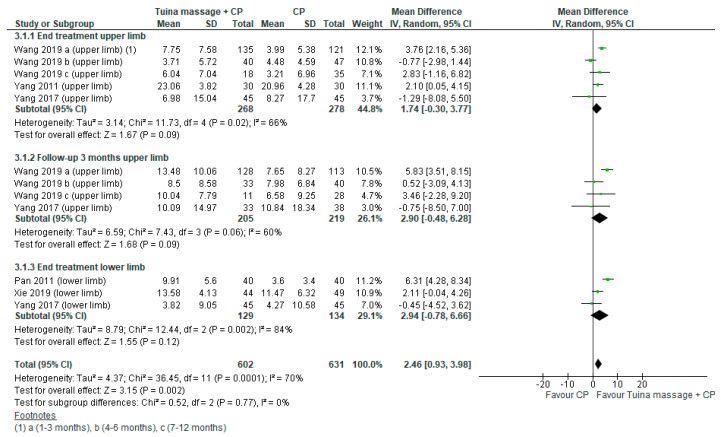
Forest plot of comparison Tuina massage plus conventional physiotherapy (CP) versus CP for motor function evaluated by Fugl Meyer Assessment Scale.

**Figure 3 ijerph-18-04424-f003:**
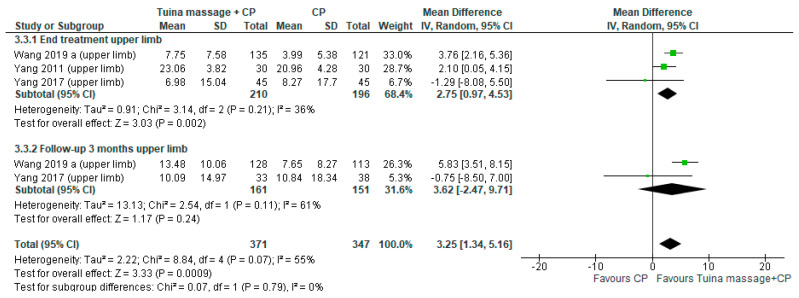
Forest plot of comparison Tuina massage plus conventional physiotherapy (CP) versus CP for upper limb motor function in subacute stage (≤3 months) evaluated by Fugl Meyer Assessment.

**Figure 4 ijerph-18-04424-f004:**
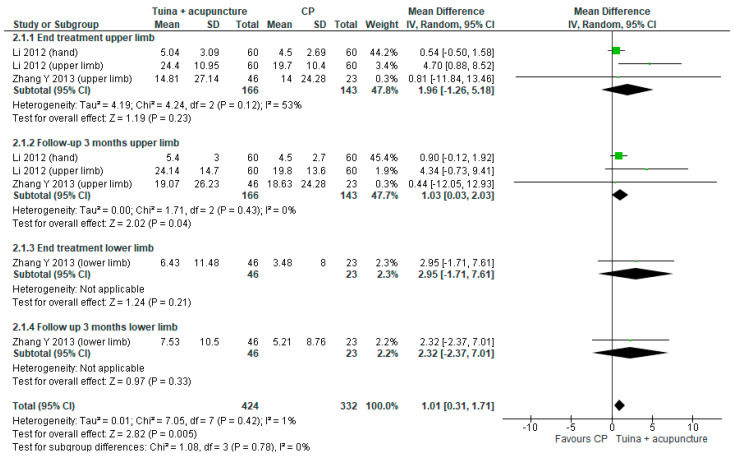
Forest plot of comparison Tuina massage plus acupuncture versus conventional physiotherapy (CP) for upper/lower limbs motor function evaluated by Fugl Meyer Assessment Scale.

**Figure 5 ijerph-18-04424-f005:**
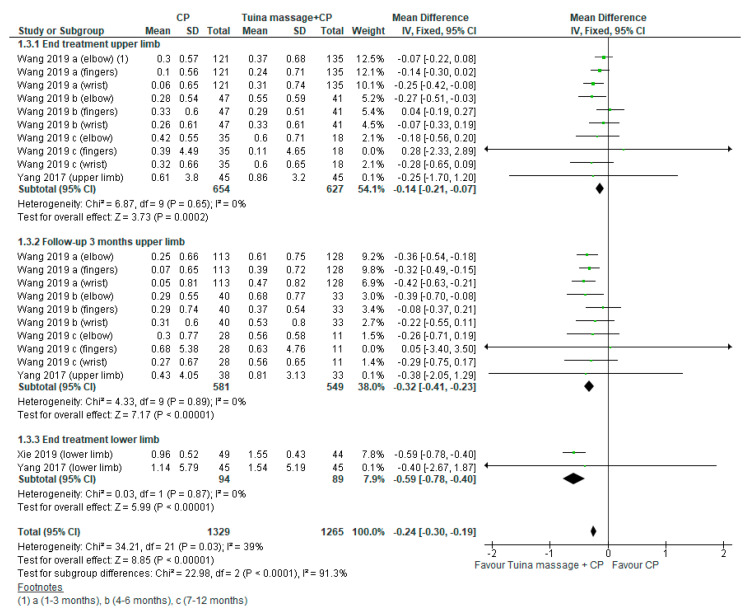
Forest plot of comparison Tuina massage plus conventional physiotherapy (CP) versus CP for upper/lower limbs spasticity evaluated by modified Ashworth Scale.

**Figure 6 ijerph-18-04424-f006:**
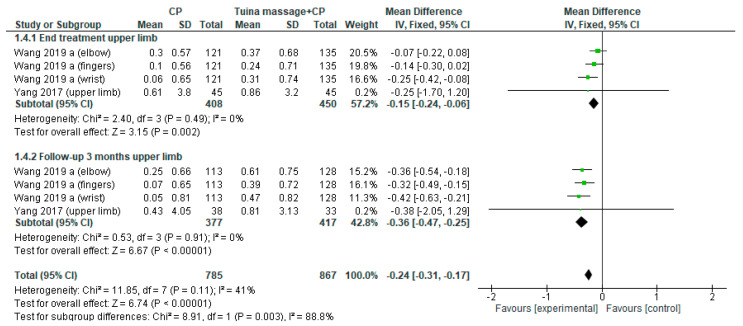
Forest plot of comparison Tuina massage plus conventional physiotherapy (CP) versus CP for upper limb spasticity in subacute stage (≤3 months) evaluated by modified Ashworth Scale.

**Figure 7 ijerph-18-04424-f007:**
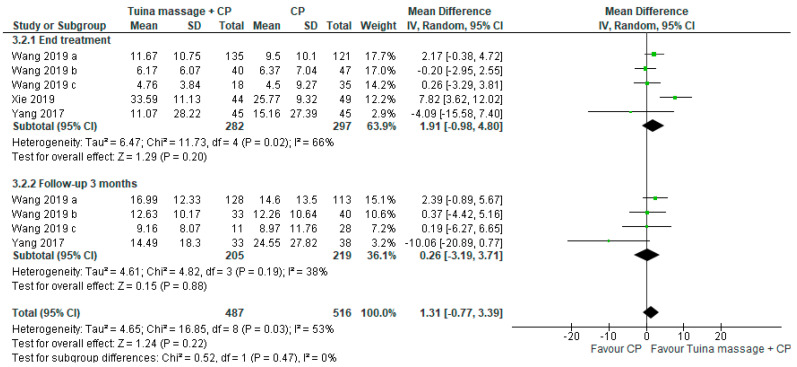
Forest plot of comparison of Tuina massage plus conventional physiotherapy (CP) versus CP for activities of daily living assessed by modified Barthel Index.

**Table 1 ijerph-18-04424-t001:** General overview of selected studies and patient’s characteristics.

Author, Year, Country	Groups (n)	Age (Years) Mean (SD)	Gender (Male/ Female)	Stroke Type (Ischemic/ Hemorrhagic)	Time Since Stroke Mean (SD)	Affected Side (Right/Left/ Bilateral)	Stroke Severity Mean (SD)
Mok, 2004 [28] China	EG:51 CG:51	EG:73.3 ± 6.63 CG:73.1 ± 6.64	51/51	NR	NR	NR	NR
Jin-su, 2005, [29] China	EG:52 CG:40	Ranged from 32 to 86 y Mean age 62.7	61/31	NR	From 2 h. to 24 d.	NR	MESSS scale: EG: mild (10), moderate (25), severe (17) CG: mild (8), moderate (23), severe (11)
Amanullah, 2011 [38] India	EG:20 CG:20	EG:55.57 ± 11.56 CG:54.30 ± 11.99	36/4	100% ischemic	Stroke onset between 4 w. to 5 y	13/ 27	NR
Pan, 2011 [39] China	EG:40 CG:40	53.65 ± 7.82	EG:28/12 CG:25/11	100% ischemic	EG:41.1 ± 12.9 d. CG:42.9 ± 13.2 d.	EG:24/16 CG:22/18	EG:55.31 ± 7.54 CG:54.68 ± 8.21
Yang, 2011 [40] China	EG:30 CG:30	EG:58.15 ± 10.26 CG:57.98 ± 10.91	EG:20/11 CG:19/11	EG:14/16 CG:17/13	EG:35.95 ± 10.02 d CG:36.21 ± 9.59 d	EG:17/13 CG:18/12	NR
Zarnigar, 2012 [41] India	EGa:20 EGb:20 CG:20	Ranged from 50 to 59 19 patients	50/20	NR	NR	NR	NR
Li, 2012 [42] China	EG:60 CG:60	EG:62 ± 12 CG:61 ± 13	EG:40/20 CG:41/19	100% ischemic	EG:28 ± 6 d CG:27 ± 5 d	EG:28/32 CG:24/36	NR
Wu, 2013 [43] China	EG:60 CG:60	EG:60.1 ± 10 CG:62.7 ± 11.3	EG:33/27 CG:32/28	EG:22/38 CG:24/36	NR	NR	NR
Zhang X, [44] 2013 China	MG:30 AG:30	MG:63.7 ± 7 AG:64 ± 7	MG:22/8 AG:20/10	NR	MG:52 ± 13 d AG:52 ± 15 d	NR	NR
Zhang Y, [45] 2013 China	EG:46 CG:23	EG:65.74 ± 10.28 CG:66.95 ± 10.95	EG:33/9 CG:13/6	100% ischemic	EG:38.00 ± 40.04 h CG:31.79 ± 37.38 h	NR	NIHSS scale EG:8.36 ± 3.03 CG:7.68 ± 2.38
Thanakiatpinyo, 2014 [30] Thailand	EG:24 CG:26	EG:60.0 ± 6.9 CG:65.8 ± 8.1	EG:22/2 CG:15/11	EG:12/12 CG:15/12	onset ≥ 3 m	NR	NR
Han, 2015 [31] China	EG:110 CG:110	EG:51.2 ± 2.1 CG:52.8 ± 1.7	EG: 56/54 CG: 58/52	EG:69/41 CG:72/38	EG:13.3 ± 5.2 d CG:13.9 ± 4.7 d	NR	NR
Ahmed, 2015 [32] India	EG:20 CG:20	From 18–70 y	NR	100% ischemic	From 4 w to 5 y	NR	NR
Di, 2017 [33] China	EG:75 CG:75	EG:61.4 ± 5.2 CG:61.7 ± 5.3	EG:48/27 CG:46/29	EG:42/33 CG:39/36	≤3 m	NR	NR
Lee, 2017 [34] Korea	EG:7 CG:7	EG: 64.3 ± 2.2 CG: 65.0 ± 5.0	EG:4/3 CG:4/3	EG:3/4 CG:3/4	EG:13.0 ± 3.1 m CG:13.6 ± 1.4 m	NR	NR
Yang, 2017 [35] China	EG:45 CG:45	EG:59.93 ± 16.87 CG:62.73 ± 11.22	EG:34/11 CG:27/18	NR	EG:3.0 ± 5.0 m CG:3.0 ± 7.0 m	EG:21/26 CG:24/19	NR
Wang, 2019 [36] China	EG:193 CG:204	a:EG:55.57 ± 11.56 CG:54.30 ± 11.99 b:EG:57.80 ± 11.34 CG:51.43 ± 13.07 c:EG:52.90 ± 12.89 GC:57.17 ± 10.97	a:EG: 99/44 CG: 93/34 b:EG: 35/13 GC: 40/13 c:EG: 20/6 CG:29/12	a:EG:96/47 CG:85/42 b:EG:32/16 CG:35/18 c:EG:17/9 CG:27/14	a:CG:1.69 ± 0.77 CG:1.70 ± 0.76 b:EG:4.84 ± 0.83 CG:4.77 ± 0.78 c:EG: 9.24 ± 1.43 CG:9.42 ± 1.53	a:EG:61/62/20 CG:55/54/18 b:EG:20/21/7 CG:22/23/8 c:EG:12/11/3 CG:17/17/7	NR
Xie,2019 [37] China	EG:44 CG:49	EG:51.6 ± 4.9 CG:52.48 ± 5.4	EG:27/17 CG:29/20	EG:24/20 CG:26/23	EG:43.9 ± 20.4 d CG:44.3 ± 21.4 d	GE:19/25 CG:22/27	NR

AG: acupuncture group, CG: control group, d: days, EG: experimental group, h: hours, m: months, MG: massage group, NIHSS: National Institutes of Health Stroke Scale, NR: Not reported, SD: standard deviation, w: weeks, Wang 2019 (a: 1–3 months, b: 4–6 months, c: 7–12 months), y: years.

**Table 2 ijerph-18-04424-t002:** Assessment of outcomes and results.

Author, Year	Evaluation			Groups	Results
	Outcome	Tool	Period		
Mok, 2004	Pain perception shoulder Anxiety level Blood pressure Heart rate Patients’massage perceptions	VAS Stai Dinamap Monitor Questionnaire	T0: Before treatment T1: After treatment T2: 3 days Follow-up	EG: Slow-stroke back massage CG:CP	Pain, anxiety, blood pressure, heart rate Better EG after treatment and follow-up
Jin-su, 2005	Stroke disability	MESSS scale	T0: Before treatment T1: After treatment	EG: Tuina massage + acupuncture CG: Western medicine (drugs)	Neurologic impairment degree Better EG
Amanullah, 2011	Motor function upper/lower limb	STREAM	T0: Before treatment T1:15th day T2: 0th day T3: After treatment	EG: Dalk massage with Roghan Seer CG: Sham Dalk massage with petroleum jelly	Voluntary movement lower limb and basic mobility Better EG after treatment Voluntary movement of upper limb *
Pan, 2011	Motor function lower limb Gait Gait analysis	FMA-L 10-MWT Motion Analysis	T0: Before treatment T1: After treatment	EG: Tuina massage + CP CG:CP	FMA-L Better EG 10 MWT Better EG Gait analysis Better EG
Yang, 2011	Motor function upper limb Pain shoulder-hand Edema	FMA-U VAS 4-point Likert scale	T0: Before treatment T1: After treatment	EG: Tuina massage + CP CG:CP	FMA-U, VAS and edema Better EG after treatment
Zarnigar, 2012	Motor function upper limb Stroke disability Gait	FMA-U FIM 10-MWT	T0: Before treatment T1: After treatment	EGa: Dalk Massage EGb: Exercises CG: Unani medicine drugs	FMA-U Better EGa after treatment FMI Better EGb 10 MWT Better EGb
Li, 2012	Motor function upper limb/ hand Pain shoulder-hand Stage of shoulder-hand syndrome Stroke disability	FMA-U FMA-H NPRS Steinbrocker classification mRS	T0: Before treatment T1: After treatment T2: 3 months Follow-up	EG: Tuina massage + electrical acupuncture CG:NDT + PNF + scapular mobilizations	NPRS scores shoulder passive movement 90°, Stage of shoulder-hand syndrome and mRS Better EG after treatment and follow-up FMA-U Better EG after treatment FMA-H *
Wu, 2013	Balance	FM-B	T0: Before treatment T1: After treatment	EG: Tuina massage + balance training CG: Balance training	FM-B Better after treatment EG
Zhang X, 2013	Motor function upper limb ADL Stroke disability Quality of life	FMA-U BI mRS SS-QOL	T0: Before treatment T1: After treatment	EG: Tuina massage + CP CG: Acupuncture + CP	FMA-U, BI, mRS and SS-QOL *
Zhang Y, 2013	Motor function upper/lower limb Stroke severity ADL Stroke disability	FMA-U FMA-L NIHSS BI mRS	T0: Before treatment T1: After treatment T2: 3 months Follow-up	EG: Tuina massage + acupuncture CG: CP	FMA lower limb and NIHSS Better EG after treatment and follow-up FMA upper limb * BI * mRS *
Thanakiatpinyo, 2014	Spasticity elbow and knee ADL Anxiety and Depression QoL	MAS BI HADS Pictorial Thai QoL test	T0: Before treatment T1: After treatment	EG: Thai massage CG:CP	Trends in decreasing spasticity, anxiety, and depression scores but no difference between groups ADL * QoL *
Han, 2015	Motor function upper limb Spasticity upper limb ADL	FMA-U MAS mBI	T0: Before treatment T1: After treatment	EG: Tuina massage + medicinal herbs (oral) CG:CP	MAS, mBI and FMA-U Better EG
Ahmed, 2015	Motor function upper/lower limb	STREAM	T0: Before treatment T1: After treatment	EG: Dalk massage with Roghan Malkangani + Unani medicine drugs CG: Western medicine (drugs)	Voluntary movement upper/lower limb and basic mobility Better EG
Di, 2017	Spasticity upper limb Stroke severity	MAS NIHSS	T0: Before treatment T1: After treatment	EG: Tuina massage CG:CP	MAS and NIHSS Better EG
Lee, 2017	Anxiety Mood state Sleep satisfaction Body temperature	4-point Likert scale MAACL VAS Infrared thermography	T0: Before treatment T1: After treatment	EG: Swedish massage + foot bath + CP CG:CP	4-point Likert scale, MAACL, VAS Infrared thermography Better EG after treatment
Yang, 2017	Motor function upper limb Spasticity upper limb ADL	FMA-U MAS mBI	T0: Before treatment T1: After treatment T2:Follow-up 3 months	EG: Tuina massage + CP CG: Sham Tuina Massage + CP	MAS elbow flexors, wrist flexors, knee flexors, knee extensors. Better EG after treatment and 3 months follow-up FMA-U * mBI *
Wang, 2019	Motor function upper/lower limbs Spasticity elbow, wrist and finger flexors ADL	FMA-Total FMA-U MAS mBI	T0: Before treatment T1: After treatment T2: Follow-up 3 months T3: Follow-up 6 months	EG: Tuina massage + CP CG:CP	MAS elbow, wrist and fingers flexors Better EG within 1-3 weeks after stroke onset and 3-6 months’ follow-up FMA upper limb Better EG after treatment and 3-6 months’ follow-up ADL *
Xie, 2019	Motor function lower limb Spasticity lower limb ADL	FMA-L MAS mBI	T0: Before treatment T1: After treatment	EG: Tuina massage + CP CG: CP	MAS lower limb, FMA-L and mBI Better EG after treatment ADL *

ADL: Activities of daily living, BI: Barthel Index, CG: Control group, CP: Conventional physiotherapy, EG: Experimental group, FMA: Fugl-Meyer Assessment, FM-B: Balance Subscale of the Fugl-Meyer Test, FIM: Functional Independence Measure, HADS: Hospital Anxiety and Depression Scale, HAMD: Hamilton depression scale, MACCL: Multiple Affect Adjective Check List, MAS: Modified Ashworth Scale, mRS: modified Rankin Scale, mBI: modified Barthel Index, MESS: Mangled Extremity Severity Score, 10-MWT: 10 metres walking test, NDT: Neuro-developmental treatment, NIHSS: National Institutes of Health Stroke Scale, NPRS: Numeric pain rating scale, PNF: Proprioceptive neuromuscular facilitation, QoL: Quality of life, RMS: surface electromyogram root mean square value, SF-HSTC: Health scale of traditional Chinese medicine, SS-QOL: Stroke-specific quality of life scale, STAI: State-Trait Anxiety Inventory, STREAM: Stroke Rehabilitation Assessment of Movement, VAS: Visual Analogue Scale. * Not statistically significant differences between groups.

## Data Availability

Not applicable.

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
