# Peer review of "The Effectiveness of Massage Therapy for Improving Sequelae in Post-Stroke Survivors. A Systematic Review and Meta-Analysis"

_ijerph, 2021, doi:10.3390/ijerph18094424_

Round 1
Reviewer 1 Report
The authors have undoubtedly put a great deal of work and effort into this review, but in the process I feel that they need to refine their work in order to make it a more succinct article for publication.
The article is too long with too many tables, many of which I have found confusing. After several reads I believe I know what the authors were trying to say BUT in order to publish such a review it needs to undergo some rather drastic editing.
May I also suggest that since in the end the article looked at the effects of a specific massage (Tuina) on the sequelae of stroke patients, perhaps this should be the focus of your article. I also feel that with a subject such as massage the grey literature may contain some interesting material and thus by limiting yourselves to systematic reviews only you might miss some highly relevant literature. Obviously if you do include articles other than RCTs you would need to find other critical appraisal tools.
I do believe that what you are trying to add to the body of literature on the treatment of stroke patients does have an importance place - but your article needs a great deal of editing and rewriting in order to make it publishable.

Author Response
First of all, thanks you for your comments and suggestions that allowed us to greatly improve the quality of the manuscript.
Title: We changed “sequeals” by “sequelae”
Abstract: We were changed “A systematic review of the most important medical databases up to December 2020” by “A systematic review of the nine medical databases up to December 2020”
Introduction: We added Physical therapy “may” contribute
We done the change “atletes” by “athletes”
2.2. Eligibility Criteria:
We chose randomized controlled trial (RCT) because we performed a meta-analysis. We think this is where the true power of the methodology comes into play. A meta-analysis of RCT is the highest level of evidence. According SUPPLEMENTARY GUIDANCE FOR AUTHORS UNDERTAKING REVIEWS WITH THE COCHRANE CONSUMERS AND COMMUNICATION REVIEW GROUP. These documents are available online at http://cccrg.cochrane.org/author-resources. it is possible to prepare a systematic review that includes a slightly broader range of experimental study designs than RCTs alone. This might be appropriate, for example, in situations where few RCTs have been conducted or where it is not possible (for ethical or practical reasons) to randomize people to receive an intervention. In our study we included 18 RCTs, it is enough for evaluating the effectiveness of therapeutic massage. Meta-analyses are a key component of evidence-based health care. Such analyses pool individual RCTs together to arrive at an overall estimate of the effect of the intervention under consideration. Meta-analyses offer several potential advantages. They provide a systematic and explicit method for synthesizing evidence, a quantitative overall estimate (and confidence intervals) derived from the individual studies, and early evidence as to the effectiveness of treatments, thus reducing the need for continued study.
We changed the date “published between January 1961 and December 2020”
This was removed “and therapeutic massage as part of a program in which isolating the effects of massage was possible”.
2.3. Data Items
The search strategy was formulated using a PICO framework. (P) human adult post-stroke survivors (I) receiving therapeutic massage alone, or in addition to another rehabilitation approach (C) compared to subjects receiving conventional physiotherapy or another rehabilitation approach; and (O) changes analysis of motor function, spasticity, activities of daily living, pain, anxiety, and quality of life with or without follow-up post-treatment.
The primary outcome was to determine if massage therapy is effective to improve motor function and to reduce spasticity in post stroke survivors. The secondary outcomes were: to determine if massage therapy is effective in reduce pain, to improve balance, gait, anxiety, activities of daily living and quality of life and to identify any adverse effects of massage.
We re-written these first and second paragraphs by
“The search strategy was designed using the PICO framework. (P) Adult post-stroke survivors, (I) receiving therapeutic massage alone, or combined with another rehabilitation approach or conventional physiotherapy, (C ) compared to patients receiving another rehabilitation approach or conventional physiotherapy and (O) changes in the analysis of motor function, spasticity, activities of daily living, anxiety, pain, balance, gait, stroke disability and quality of life with or without a follow-up assessment."
"The primary outcomes were upper/lower limbs motor function and spasticity. The secondary outcomes were activities of daily living, pain, anxiety, balance, gait, stroke disability and quality of life. We also added as a secondary outcome to identify any adverse effects of the therapeutic massage".
Search:
The mean value and standard deviation were removed.
This phrase was changed: "Conventional physiotherapy intervention was described."
PRISMA diagram was changed with more explanations.
Table 2: we removed "setting" column and "first ever stroke" column.
Table 4: we changed “application” by “application”, “affected” by “paretic”, “discription” by “description”.
Discussion:
We removed these phrases, probably are not relevant: "The other branches are acupuncture, Chinese herbal medicines and exercise that are regularly administered to patients who have suffered from stroke, and they are usually used in combination. The application of Chinese Medicine in stroke patients is widely accepted and increasing".
This phrase was not removed because it is relevant to understand the effect of Tuina in spasticity. “Yin-Yang imbalance also contributes to upper and lower limb spasticity following stroke, manifesting as ‘flaccidity of Yang and spasm of Yin’ [76].”

Reviewer 2 Report
Thank you for recommending me as a reviewer. This meta-study was to evaluated the effectiveness of therapeutic massage to improve the sequelae of stroke survivors. The results of this study are interesting and of high academic value. Meta research was well done according to PRISMA's procedures. If the authors complete minor revisions, the quality of the study will be further improved.
- The introduction section is well written. However, the need for meta-research on "therapeutic massage" is insufficient. For example, the author can write, "Because the trends of previous studies on therapeutic massage' were reported heterogeneously, we performed a meta-study for evidence-based treatment."
2. The methods section is well written.
3. page 15: In this paper, expressions such as "p = 0.00001" are repeated for the significance level, but it would be better to unify it as "p<0.001".
4. page 14-15: It would be a good idea to unify the number of decimal places for the significance level in the results section.
Author Response
First of all, thank you for your comments and suggestions.
In the introduction section we have added “Because the trends of previous studies on therapeutic massage were reported heterogeneously, we performed a meta-analysis and systematic review for evidence-based treatment."
We have changed "p = 0.00001" by "p<0.001".
page 14-15: It would be a good idea to unify the number of decimal places for the significance level in the results section. It was done.

Round 2
Reviewer 1 Report
Although this paper is greatly improved there are still a number of changes I believe that are required before it is publishable.
I have made a number of comments in the first sections of this paper - but I still feel that the number of tables and amount of information given in them is excessive. No one is going to read all that information in a published article - you MUST precis it and if you feel that someone somewhere may want to read it - then add it as supplementary pages.
Again I am aware of the vast amount of work you have put into this and no doubt you are passionate about the addition of massage to the treatment regimen of stroke patients - BUT less is more. Please condense and then maybe you will have a useful addition to the literature.
One last thing - numbers in the thousands are not written with a full stop.

Author Response
Many thanks you for your comments and suggestions.
We have made these changes:
Abstract: This was added “from January 1961 to”,” meta-analysis of RCTs”
We have changed numbers in the thousands 3,196.
Introduction
This phrase “Only one study performed Swedish massage” has been removed.
Introduction:
We removed the comma agonist/antagonist, “agonist antagonist”.
This phrase “Together, they result in impairments and functional problems that can predispose to costly complications” has been changed by “They involve together in impairments and functional problems that can lead to costly complications”.
We added Parkinson’s disease
We have changed this phrase “To our knowledge, only one meta-analysis about post-stroke constipation has been published” by “ and in post-stroke constipation [17]”.
2.2 Eligibility criteria
This phrase was rewritten “Since only RCTs were to be analysed all other forms of literature were excluded as comments, reviews, observational studies, books, poster/oral abstract communications, case reports, non-randomized studies, protocols, systematic reviews and practice guidelines”.
This phrase was removed “The following interventions were excluded: therapeutic massage administered using devices such a vibrator”
This phrase “We also added as a secondary outcome to identify any adverse effects of the therapeutic massage” was changed by a heading.
In our opinion PRISMA diagram should be in results section. The PRISMA diagram was changed.
We are rewritten Table 1 and table 2 and table 4 is in supplemental material.